# Interplay Between Glutamine Metabolism and Other Cellular Pathways: A Promising Hub in the Treatment of HNSCC

**DOI:** 10.3390/cells14241962

**Published:** 2025-12-10

**Authors:** Teresa Stefania Dell’Endice, Francesca Posa, Giuseppina Storlino, Lorenzo Sanesi, Lucio Lo Russo, Giorgio Mori

**Affiliations:** 1Department of Clinical and Experimental Medicine, University of Foggia, Viale Pinto 1, 71122 Foggia, Italy; stefania.dellendice@unifg.it (T.S.D.); giuseppina.storlino@unifg.it (G.S.); lucio.lorusso@unifg.it (L.L.R.); giorgio.mori@unifg.it (G.M.); 2Department of Medicine and Surgery, Kore University of Enna, 94100 Enna, Italy; lorenzo.sanesi@unikore.it

**Keywords:** head and neck cancer, head and neck squamous cell carcinoma, glutamine metabolism, targeted therapy, precision medicine in oncology, ferroptosis

## Abstract

**Highlights:**

**What are the main findings?**
HNSCC tumors are highly dependent on glutamine (Gln) metabolism, with GLS1 as a key player in driving tumorigenesis.Targeting GLS1, ASCT2, and c-Myc in HNSCC disrupts Gln metabolism, with promising antitumor effects and potential immune benefits.

**What is the implication of the main finding?**
Metabolic rewiring reduces the efficacy of single-agent therapies, highlighting the need for rational combination strategies.Tumor heterogeneity, model limitations, and lack of predictive biomarkers hinder clinical translation and require improved patient stratification.

**Abstract:**

Head and neck squamous cell carcinoma (HNSCC) is the most common and aggressive histologic subtype of head and neck cancer (HNC), difficult to treat effectively. Here, we discuss several studies on human and mouse HNSCC cell lines arising from the mucosal epithelium of various anatomical sites, as well as recent studies using murine models, focused on targeting key checkpoints in the glutamine (Gln) metabolism pathway, either alone or in synergy with other signaling pathways, as a potential therapeutic strategy for HNSCC. Emerging evidence demonstrates a complex interplay between Gln metabolism and pathways mediating altered cellular mechanisms, including ferroptosis, immune system evasion, mitochondrial energy production, and oncogenic transcriptional control. This review examines currently available gene expression databases and protein expression analyses of Gln metabolism-related components in tissue samples from HNSCC patients. From a translational perspective, the co-administration of pharmaceutical agents and biologic products targeting distinct molecular pathways, integrated with radiotherapy (RT) or chemotherapy (CT), may produce superior anti-HNSCC efficacy, thereby improving clinical outcomes and extending patient survival. Multimodal strategies represent a key direction in precision oncology, enabling personalized therapeutic interventions to suppress metastatic dissemination and disease progression more effectively. Therefore, an integrated therapeutic approach represents a promising path to defeat HNSCC.

## 1. Introduction

Malignancies arising from the upper aerodigestive tract fall into the group of head and neck cancer (HNC), classified according to their anatomical sites of origin (oral cavity, lips, pharynx, nasopharynx, oropharynx, hypopharynx, larynx, salivary glands, thyroid, nasal cavities and paranasal sinuses) [1,2].

Over 90% of HNCs are represented by head and neck squamous cell carcinoma (HNSCC), which develops from the mucosal epithelium lining the oral cavity, pharynx and larynx [1,2]. According to the latest 2022 GLOBOCAN database, HNSCC remains the seventh most common malignant neoplasm worldwide. Major risk factors include tobacco and alcohol use [1,3], while the incidence of oropharyngeal squamous cell carcinoma (OPSCC) related to human papillomavirus (HPV), mostly HPV-16 variant, has been increasing in recent years [1,4]. Clinically, the routine approach to treat HNSCC consists of surgical resection followed by radiation and/or adjuvant chemotherapy (CT) [5,6].

However, despite different therapeutic advancements, HNSCC remains aggressive, with a poor prognosis, mainly due to late diagnosis, frequent metastasis, relapse, intrinsic drug resistance, and treatment-associated toxicity. Hence, the development of new biomarkers for early detection and screening, along with the development of targeted therapies, represents a critical priority and is necessary also for advancing precision medicine.

Growing evidence highlights how alterations in metabolic processes are relevant in HNSCC. A significant metabolic reprogramming occurs in HNSCC cells, which supports uncontrolled cell growth, survival, and invasion. Among the metabolic pathways involved, glutamine (Gln) metabolism plays a pivotal role in the pathophysiology of HNSCC [7] and is associated with tumor progression [8]. The expression of glutaminase (GLS1)—the key enzyme that converts Gln into glutamate through glutaminolysis—is significantly upregulated in both primary and metastatic HNSCC tissues [9] and its expression correlates positively with advanced clinicopathological parameters [10]. Consistently, metabolomic analyses revealed markedly elevated glutamate levels in primary HNSCC tissues [2,11], with even higher levels detected in metastatic tissues, compared with adjacent normal pharyngeal mucosa [8,9]. Increased glutamate levels have also been detected in both saliva and plasma samples of HNSCC patients [12].

In contrast, Gln concentrations are lower in primary tumors and further reduced in metastatic HNSCC tissues [12], suggesting overactivation of glutaminolysis and confirming a strong reliance of HNSCC on Gln.

This metabolic reprogramming leads to the accumulation of high extracellular glutamate levels, which may influence tumor biology through activation of metabotropic (mGluR1-8) and ionotropic (iGluR) glutamate receptors [13,14]. Indeed, glutamate has been shown to act as an autocrine and paracrine signaling molecule within the tumor microenvironment (TME).

Numerous studies reported the involvement of both classes of glutamate receptors in the carcinogenesis of multiple tumor types, including neuronal, breast, kidney, prostate, colorectal, gastrointestinal, melanoma, ovary, and oral squamous cell carcinoma (OSCC) [13,14,15]. Supporting this evidence, OSCC samples exhibited strong mGluR5 immunostaining expression [16], with a significant association with advanced tumor stages. Furthermore, high mGluR5 expression correlated with a significantly decreased 5-year survival rate, suggesting a relevant implication in OSCC tumor progression [16]. In addition, Choi et al. demonstrated that the ionotropic N-methyl-D-aspartate (NMDA) receptor 1 is overexpressed in OSCC and is significantly associated with lymph node metastasis, tumor size, advanced cancer stage, and poorer patient survival [17]. Notably, the NR1 subunit was found to be involved in post-transcriptional regulatory processes. Finally, GRM5 methylation has been reported to increase the risk of OSCC, although the association did not reach statistical significance and may involve epigenetic mechanisms [18].

Taken together, these findings highlight the central role of the glutamine-glutamate axis in HNSCC development and progression, underscoring its potential as a promising therapeutic target. Current treatments for HNSCC do not exploit tumor cells’ dependence on Gln, despite this dependence being increasingly recognized as a cause of therapeutic failure by promoting radioresistance, chemoresistance, and immune evasion. Therefore, metabolic interventions targeting the Gln-glutamate axis represent a clear, unmet clinical need. The present review summarizes recent in vitro, in vivo, and ex vivo studies investigating mainly Gln catabolism in HNSCC and its putative interplay with other signaling pathways, with the aim of identifying new and effective therapeutic strategies for HNSCC.

## 2. Clinical Management of HNSCC: Treatment Toxicities and the Emerging Role of Glutamine Supplementation

The treatment of choice for HNSCC, as previously mentioned, is surgical removal followed by radiotherapy (RT), CT, or chemoradiotherapy (CRT). RT and CRT are often associated with clinically significant toxicities that can negatively impact therapeutic outcomes. One of the most debilitating adverse effects is oral mucositis (OM), characterized by mucosal thickening, ulcer formation, painful swallowing and the consequent need to interrupt ongoing therapies. The average incidence of OM in HNC patients is around 80–90%, with severe forms occurring in 50 to 60% of cases [19,20].

The pathogenesis of OM is complex, and there is currently no universally accepted standard of care; nevertheless, Gln appears to play a key role in this scenario. Although cancer therapy can cause Gln deficiency in patients [21], raising concerns that supplementation could promote tumor growth, clinical studies have demonstrated that Gln supplementation during therapies does not increase tumor growth or worsen oncological outcomes.

Several studies have shown that oral Gln supplementation in HNSCC patients delays the onset of chemoradiation-induced OM, reduces its severity and duration, with a concomitant decreased use of opioid analgesics, nasogastric tube feeding, and therapy interruptions [21,22]. Moreover, recent analyses have investigated whether Gln supplementation affects survival or oncological efficacy over a 5-year follow-up. This study did not find significant differences in patient overall survival (OS) compared to controls [12]. However, it should be considered that these results were obtained on a small cohort of 40 patients. Therefore, larger multicenter RCTs are needed to validate these preliminary findings. Overall, Gln supplementation may represent a beneficial strategy for managing OM in HNSCC, but these benefits may not be sufficient to influence survival outcomes. Although OM is frequent in HNSCC, not all patients develop severe forms requiring treatment interruption, and this partly limits the measurable impact of supplementation on long-term outcomes. Moreover, reducing OM improves treatment tolerance and patient comfort but does not directly modify tumor biology, which explains why OS remains unchanged. In small cohorts, such as those included in the cited trials, these effects tend to be further diluted.

We can deduce that Gln can be safely implemented without impacting the effectiveness of cancer therapy and preventing the mucosal epithelium from oxidative stress by increasing the intracellular levels of glutathione (GSH), in normal tissues. GSH plays an important function in detoxification, cell signaling, metabolism, gene expression, and immune system function.

In addition to standard care guidelines, Cetuximab, a monoclonal antibody targeting the Epidermal Growth Factor Receptor (EGFR) [23], and Nivolumab and Pembrolizumab, two immunotherapeutics against PD-1 (programmed cell death protein 1), have been approved for HNSCC treatment [24,25,26,27].

The benefits of Cetuximab are further enhanced when combined with platinum-based treatment, leading to a longer median OS [24,25,26,27,28].

Nevertheless, patients with HPV-related HNSCC are more sensitive to RT, CT [29] and immunotherapy [30] showing a more favorable prognosis [29], improved OS, and significantly lower incidence rates of second primary tumors and metastases than patients (HPV−) [3].

Recently, multiomics analysis has identified numerous genes highly expressed in HNSCC tissue tissues, at both the mRNA and protein levels, which may represent potential biomarkers and therapeutic targets [31]. Ongoing clinical trials for HNSCC include exploration of HMBD-001 (antibody against HER3), Lenvatinib (antibody against multiple tyrosine kinases receptors) [32], therapeutic vaccines against HPV-related antigens, adoptive cell therapy and immunotherapies.

Several preclinical studies and clinical trials are also developing small-molecule inhibitors targeting signaling pathways such as: PI3K/AKT/mTOR, EGFR, VEGF, FGFR, MEK/ERK, MET, CDK4/6, and Notch signaling to enhance radiosensitivity [6,33]. These regimens, when combined with CT or RT, can minimize adverse effects and improve patient safety. Moreover, only a few drug screening studies have used patient-derived tumoroids to mimic the in vivo microenvironment and predict treatment responses, thereby supporting precision oncology [34].

Thus, identifying new effective therapies to overcome HNSCC requires investigating the complex interactions among the TME, the immune evasion and the dysregulated metabolism. Over the last few decades, several studies have shown that various metabolic signaling pathways are altered and reprogrammed in HNC, as well as in other tumor types, to support the rapid growth and progression, thereby interfering with therapy success. Based on the metabolic profile, using The Cancer Genome Atlas (TCGA) and Gene Expression Omnibus (GEO) databases, four HNSCC subtypes were identified: glycolytic, cholesterogenic, quiescent and mixed [35]. These metabolic subtypes were then correlated with genomic alterations, revealing TP53, CDKN2A, PIK3CA, LRP1B and FLG as the most frequently mutated genes [35].

## 3. Glutamine Signaling Pathway and Its Roles in Cellular Functions

In order to proliferate, mammalian cells need to synthesize nucleotides, proteins, and lipids by importing nutrients from the extracellular microenvironment. Glucose and Gln, a non-essential amino acid (NEAA), are the two principal exogenous carbon and nitrogen sources imported from bloodstream into the cells to support cell survival and macromolecule biosynthesis [13]. The uptake of Gln from the extracellular environment into cells, including tumor cells, is mediated via the plasma membrane transporters such as alanine, serine, cysteine-preferring transporter 2 (ASCT2), a Na^+^-dependent neutral amino acid transporter encoded by solute-linked carrier family A1 member 5 (SLC1A5) [14,16]. Gln is mainly synthesized in skeletal muscle by the ubiquitous mitochondrial enzyme Gln synthetase (GLUL), from glutamate and ammonia (NH_3_), in the cytosol. Small amounts of Gln are also released into the blood circulation by the liver, lungs, and brain and subsequently utilized by multiple tissues [10]. Cytosolic Gln is transported in the mitochondria, across the inner mitochondrial membrane via the SLC1A5 variant transporter, where it is hydrolyzed by GLS to ammonium-ion (NH_4_^+^) and glutamate [36], (Figure 1).

Gln hydrolysis can be catalyzed by two GLS isoforms due to differential RNA splicing, GLS1 (kidney-type glutaminase) and GLS2 (liver-type glutaminase), and both are expressed differentially across cancer types. Mitochondrial glutamate is subsequently transformed, by glutamate dehydrogenase (GDH) or transaminases (GOT2, GPT2), into α-ketoglutarate (α-KG) that enters the tricarboxylic acid (TCA) cycle, also named the Krebs cycle, to generate ATP through production of NADH and FADH2 [10]. These molecules, in turn, transfer electrons to the electron transport chain in the mitochondrial inner membrane, leading to the generation of CO_2_ and ATP by oxidative phosphorylation (OXPHOS). Glutamate translocates to the cytosolic compartment through the transporters SLC25A18 and SLC25A22, where it is recruited for the biosynthesis of GSH and NEAAs (alanine, proline, aspartate, asparagine, arginine). Citrate and α-KG, TCA intermediates, are exported into the cytosol, via SLC25A1 and SLC25A11respectively, as precursors for de novo fatty acid synthesis and NADH generation. In addition, cytosolic α-KG activates the mTORC1 (mechanistic target of rapamycin complex 1) pathway, promoting cell growth by sustaining all biosynthesis processes, and acts as a cofactor for enzymes involved in epigenetic modifications into the nucleus.

Gln catabolism leads to a high malic enzyme flux to supply the NADPH used for fatty acid synthesis. Furthermore, Gln provides carbon atoms to refill the oxaloacetate (OAA) pool. OAA is involved in several key metabolic processes, including gluconeogenesis, the urea cycle, glyoxylate cycle, amino acid synthesis, and fatty acid synthesis [37].

Thus, the TCA cycle provides intermediates that are needed as biosynthetic precursors for nucleotides, lipids, NEAAs [38] and GSH synthesis, as well as regulating numerous cellular functions and cell fate. Therefore, Gln is the major contributor to the anaplerotic replenishment of the TCA cycle as a nitrogen source and is essential for cell proliferation.

Overall, Gln uptake and its metabolism in normal cells are tightly regulated by the mTORC1 pathway and its multiple downstream effectors. Among these, the c-Myc (myelocytomatosis viral oncogene homolog) oncogene controls the activation of both GLS and SLC1A5. Simultaneously, mTORC1 signaling inhibits catabolic processes such as autophagy and mediates aerobic glycolysis thorough hypoxia-inducible factor 1-alpha (HIF-1α), a transcription factor crucial for angiogenesis and metabolic adaptation to hypoxic microenvironments. In turn, under physiological conditions, Gln availability modulates mTORC1 pathway activation [39].

The balance between mTORC1 signaling and controlled HIF-1α activity ensures that Gln utilization matches cellular energy and biosynthetic demands, preventing uncontrolled cell growth and preserving cellular functions [40,41]. Under physiological conditions, therefore, Gln utilization is regulated to adequately adapt to the actual needs of the cell. In cancer cells, this balance is altered, and various oncogenic signals constantly activate Gln transport and metabolism.

## 4. Dysregulated Glutamine Metabolism in HNSCC

Whereas normal cells require Gln to maintain their anabolic homeostasis, cancer cells are tightly dependent on Gln catabolism. This reliance persists even in the presence of other nutrients, reflecting their reprogrammed metabolic state during malignant transformation.

A peculiar feature of cancer cells is the well-known Warburg effect, which refers to their tendency to enhance glucose uptake and convert most of it into lactate to generate energy, despite the availability of oxygen [42,43]. Whereby, cancer cells prefer aerobic glycolysis with 2 ATP generated per glucose molecule, followed by lactic acid fermentation, rather than OXPHOS, which would instead have a net gain of 36 ATP molecules per glucose molecule [42]. At some point, cancer cells may also switch to glutaminolysis, converting Gln to lactate [14] via glutamate, α-KG, and the TCA cycle, to generate cellular energy, leading to a cell Gln addiction phenotype. The reliance on Gln in cancer cells is justified not only by its role as a donor of amino groups [44] or as carbon source for lactate secretion, but also because its catabolism provides high-energy NADPH cofactors, maintaining the cytosolic NADPH pool to support redox homeostasis and restore oxidized GSH [36]. Gln also plays a crucial role in mechanisms such as cellular autophagy, reactive oxygen species (ROS) stress, and the TME formation [45].

Thus, in tumors, Gln becomes a major nutrient source sustaining continuous anaplerosis and antioxidant defense. Furthermore, its demand may exceed its supply, and therefore, its uptake from the extracellular environment becomes essential. Under limited blood Gln availability cancer cells respond by the de novo synthesis of Gln by GLUL.

Increased intracellular Gln availability, elevated energy levels, and growth factors stimulation act as upstream regulators that drive the constitutive hyperactivation of mTORC1 signaling. Persistent activation of this pathway triggers and amplifies a series of downstream events that enhance Gln metabolism, thereby promoting tumor progression. A major consequence is the overexpression of c-Myc, which upregulates two key components of Gln metabolism, GLS and SLC1A5, increasing the cellular capacity for Gln utilization. Through all these changes, cancer cells potentiate Gln uptake and accelerate its conversion into glutamate, thereby fueling the TCA cycle and meeting their elevated metabolic demands.

In addition, the mTORC1 complex itself may undergo genetic or regulatory alterations that further enhance its activity, reinforcing these metabolic networks.

This metabolic rewiring enables cancer cells to survive and proliferate uncontrollably, even under hostile microenvironmental conditions such as hypoxia and nutrient deprivation. Moreover, lactate can stabilize and activate HIF-1α even in normoxic conditions, a phenomenon known as pseudohypoxia, which further promotes metabolic adaptation and malignant progression.

Indeed, scientific evidence from in vitro studies demonstrated elevated GLS protein expression levels, compared to normal human oral keratinocytes, in several human-derived HNSCC cell lines. Notable examples include FaDu (pharynx squamous cancer); Detroit 562 (pharyngeal carcinoma); CAL-27 and HN5 (both derived from tongue squamous carcinoma) [7,46]; UM-SCC-17B (supraglottis/soft tissue-neck); UM-SCC-14A (floor of mouth); HSC-3 (tongue); OSCC-3 (tongue) [9] (Table 1).

Immunohistochemical (IHC) evaluation studies of tissue sections [34] or TMA (tissue microarray) [10] from HNSCC patients—including specimens ranging from normal to metastatic stage and originating from the tongue, oral cavity, lip, larynx, and lymph-nodes—(Figure 2) displayed an upregulation of GLS enzyme expression in tumor tissues compared to normal mucosa (Table 2). Moreover, metastatic samples exhibited a higher percentage of GLS expression than primary tumors and lower Gln rate [9,10], indicating hyperactivation of glutaminolysis and its potential involvement in driving HNSCC progression. Specifically, the GLS2 isoform showed a negative correlation for with tumor grade, opposite to GLS1, suggesting distinct and antagonistic roles depending on the tumor differentiation status [10]. The Cancer Genome Atlas (TCGA) HNSCC cohort showed a mean GLS1 gene expression level approximately twice as high as that of healthy controls (Table 2). Kaplan–Meier analysis further revealed that patients in the GLS-low subgroup had significantly longer survival and disease-free periods compared to those in the GLS-high subgroup [7,10,51]. Therefore, GLS1 may be considered a potential biomarker in HNSCC and useful for selecting patients who might benefit from targeted metabolic therapies. In line with these observations, the higher GLS expression also coincides with elevated glutamate levels in metastatic HNSCC tissues, as well as in the saliva and plasma of HNSCC patients analyzed [9] (Table 2). In turn, glutamate, by mGluR and iGluR receptors, may trigger intracellular signaling pathways that promote cell proliferation and invasiveness. Altogether, these findings confirm the presence of a metabolic dysregulation in HNSCC, characterized by an adaptive shift towards Gln consumption, contributing to the aggressiveness and poor prognosis of HNSCC.

Although current evidence highlights hyperactivation of glutaminolysis and upregulation of GLS in HNSCC, it remains unclear whether these metabolic alterations are simply consequences of the tumor state or whether they directly contribute to tumor initiation and progression.

Several studies have demonstrated that in many solid tumors, such as lung adenocarcinoma, triple-negative breast cancer, and pancreatic ductal adenocarcinoma, the metabolism relies heavily on Gln to fuel their rapid growth. This dysregulation may represent both an adaptive response to the high biosynthetic demands of rapidly proliferating cancer cells and, in some contexts, a metabolic vulnerability that promotes oncogenic signaling via c-Myc, mTORC1, and redox control networks. Targeting Gln metabolism could, therefore, represent a strategy to selectively interfere with cancer cell survival without affecting normal cells.

Anyway, further research and translational studies are needed to clarify whether targeting Gln metabolism in clinical settings could effectively disrupt these survival pathways and provide therapeutic benefit.

However, because metabolic pathways are highly interconnected, there are challenges in disrupting Gln metabolism without causing significant side effects or triggering compensatory mechanisms that allow the disease to adapt.

## 5. Glutamate Receptors and Their Roles in HNSCC

Glutamate is the main neurotransmitter regulating numerous functions within the central nervous system (CNS), including those associated with neuronal disorder diseases. Its action is mediated through mGluRs and iGluRs receptors, predominantly expressed in neuronal cells. mGluRs belong to the G protein-coupled receptor (GPCR) family and are classified into three groups: Group I (mGluR1, mGluR5), in post-synaptic location with excitatory properties; Group II (mGluR2, mGluR3) and Group III (mGluR4, mGluR6, mGluR7) present in pre-synaptic sites with inhibitory function [53,54]. Specifically, mGluRs induce MAPK, PI3K/AKT, and mTOR pathways, thereby enhancing cell survival, growth, and metabolic rewiring [53,54].

Instead, iGluR, are ligand-gated ion channels and include NMDA and non-NMDA receptor [α-amino-3-hydroxy-5-methyl-4-isoxazolepropionic acid (AMPA) receptors] (iGluR1-4) and kainite (KA) subfamilies (iGluR5-7, KA1, and KA2) Kevin. Activation of NMDA and AMPA leads to increased intracellular Ca^2+^ influx, with Ca^2+^ acting as a second messenger involved in many cellular processes such as cell migration, invasion, angiogenesis, and epithelial–mesenchymal transition (EMT). Recently, preclinical and clinical evidence reported that calcium signaling modulation may also interfere with HNSCC progression [55]. According to this, it has also been shown how Ca^2+^ channels Orai-1 and Orai-2 are overexpressed in oral cancer cells and tumor tissues [56], playing a critical role in EMT and metastasis in HNSCC.

Anyway, both metabotropic and ionotropic glutamate receptors have been detected in non-neuronal cell types of peripheral organs and are widely expressed in multiple cancers, including OSCC [53,54]. In cancer cells, they may initiate oncogenic signaling cascades that promote growth, survival, and invasion [53,54]. In fact, in OSCC, mGluR5 results overexpressed [57] and may have an oncogenic role enhancing metastasis, invasion, and adhesion. The epigenetic regulation of GRM5 gene may also be involved in OSCC [18].

## 6. Glutamine Metabolism Checkpoints as a Potential Pharmacological Target in HNSCC Treatment

Given the altered Gln metabolism observed in HNSCC, it has been hypothesized that interfering with the modulation of this pathway at different key checkpoints may represent a promising strategy for treating HNSCC (Figure 1). By disrupting Gln metabolism, it may be possible to selectively deprive cancer cells of a key nutrient while minimizing effects on normal cells.

To achieve this goal, several experimental approaches in vitro, in vivo, and ex vivo have been employed to study and identify drugs acting on enzymes, metabolites, and transporters related to the Gln pathway that may potentially be applied clinically to treat HNSCC. Table 3 provides an overview of the inhibitors used and their respective mechanisms of action.

### 6.1. GLS1 Inhibition in HNSCC

Blockade of GLS1 function prevents glutamate synthesis, thereby limiting substrate availability for the TCA cycle and the production of intermediates required for the biosynthesis of macromolecules essential for cell proliferation.

Inhibition of GLS1 enzyme activity, either chemically by a Gln analog DON (6-diazo-5-oxo-L-norleucine) or via stable shRNA-mediated GLS1 knockdown in the UM-SCC-14A cell line, blocked tumorsphere formation, and led to decreased glutamate levels, impairing TCA replenishment [9]. Consistently, an oral cancer mouse model transfected with UM-SCC-14A cells knocked down for GLS1 also displayed decreased glutamate concentrations and reduced tumor volume, confirming suppression of glutaminolysis and tumorigenesis [9].

Although DON has been explored previously as a therapeutic strategy in patients with Gln-dependent cancer, its clinical use has been limited by toxicities such as stomatitis and diarrhea [33]. To overcome these limitations, a prodrug form of DON, DRP-104 (sirpiglenastat), has been developed to selectively release DON in the TME, reducing systemic toxicity. Treatment of both HPV (−) HNSCC cell lines (e.g., CAL33 and CAL27) and HPV (+) cell lines (e.g., UDSCC2 and UMSCC47) with DON or DRP104 effectively inhibited cell proliferation [33]. This response appears to be independent of HPV status, anatomical origin, or metastatic stage in these models, suggesting that Gln metabolism is a consistent vulnerability in HNSCC. Even though these findings suggest that DON or DRP-104 mediated GLS1 inhibition may suppress tumorigenesis, causal conclusions cannot be drawn from these data.

Furthermore, in both CAL33 and CAL27 cells, DON treatment impairs mTOR pathway activity thereby stimulating autophagy through modulation of ULK1, p62 and LC3B, key autophagy marker proteins [33].

This autophagic response leads to the accumulation of polyunsaturated fatty acids (PUFAs) in lipid droplets. Elevated hydrogen peroxide (H_2_O_2_) and ROS levels were also detected, promoting lipid peroxidation and ultimately driving ferroptosis [33]. Ferroptosis is an iron-dependent form of regulated cell death tightly controlled by GPX4 (selenoprotein glutathione peroxidase).

When intracellular iron levels are high, ferrous ions (Fe^2+^) participate in the Fenton reaction, converting H_2_O_2_ into highly reactive hydroxyl radicals (•OH). This, in turn, leads to an overproduction of ROS, triggering further lipid peroxidation and, ultimately, ferroptosis, as observed in cells treated with DON. Moreover, both in vitro and in vivo experiments showed that CAL27 and CAL33 cells exhibited increased GPX4 expression, as well as low levels of GSH and its oxidized form (GSSG), and increased H_2_O_2_ and ROS levels (Figure 3) following DON treatment. GPX4 normally utilizes GSH as a cofactor to convert lipid hydroperoxides into non-toxic lipid alcohols [33].

DON treatment may therefore initially sensitize cells to ferroptosis through GPX4 inactivation and ROS accumulation, followed by GPX4 overexpression, likely as a compensatory mechanism to counteract excessive oxidative stress and limit lipid peroxidation. However, these results should be interpreted with caution, and further studies on pharmacokinetics and dose–response correlation are required to better define the dose-dependent effects of DON and its prodrug DRP-104 in vivo.

Ferroptosis appears to be frequently suppressed in HNSCC. Gene expression profiles of HNSCC samples show high expression of ferroptosis suppressor protein 1 (FSP1) and related regulatory genes, with FSP1 positively correlating with GPX4, ACSL4 and FAT1 (involved in fatty acid metabolism), the antioxidant enzyme SOD2, HIF1a, and IREB2 (iron-responsive element) in HNSCC [61]. Thus, GLS1 inhibition may promote FSP1 inactivation in HNSCC cells, potentially leading to ferroptosis. However, further studies are needed to confirm a causal link between DON treatment and ferroptosis.

TCGA data confirm the correlation of high FSP1 expression with tumor progression and poorer survival [61]. A preliminary clinical study also found increased FSP1 mRNA and protein levels in tissues from cisplatin-resistant patients, with resistance closely associated with FSP1 expression [61].

These results provide evidence of a functional link between Gln metabolism and ferroptosis in HNSCC, suggesting that a dual therapeutic approach targeting both pathways in HNSCC could enhance treatment efficacy [7].

However, DON covalently binds to the active sites of several Gln-utilizing enzymes, irreversibly inhibiting their activity. Due to its lack of specificity, it may also affect other Gln-mediated signaling pathways and fatty acid metabolism as shown, leading to potential off-target effects. To minimize these risks, alternative drugs should be considered.

Selective GLS1 inhibition by the allosteric inhibitor BPTES (bis-2-(5-phenylacetamido-1,3,4-thiadiazol-2-yl) ethyl sulfide), or by treatment with Metformin, suppresses the growth of FaDu and Detroit 562 cells by inducing apoptosis and cell cycle arrest [7].

Mechanistically, BPTES inhibits glutaminolysis, reducing downstream Gln-derived metabolites, and induces G2-phase arrest by blocking cyclin E2 expression and upregulating p21. Metformin, an antihyperglycemic drug used for type II diabetes, induces G1-phase arrest by downregulating CDK1/cyclin B1 and cyclin E2 [7]. Besides its effects on the cell cycle, Metformin targets mitochondria inhibiting complex I activity in HNSCC cells [62], while also activating AMPK and suppressing mTORC1 signaling, similarly to DON effects observed in CAL27 and CAL33 HNSCC cell lines.

Both agents ultimately promote apoptosis through caspase 3 cleavage [7]. Accordingly, epidemiological and clinical data showed an association between Metformin use and reduced cancer incidence, improved prognosis, and decreased mortality in liver, colorectal, pancreatic, stomach, esophageal [7] and HNSCC cancers [60]. Therefore, the combination of BPTES and Metformin enhances the anti-neoplastic effects in FaDu and Detroit 562 cell lines, reducing cell survival, proliferation, and glucose consumption more effectively than either agent alone [7] (Figure 3).

In vivo experiments confirmed these findings: mice bearing HNSCC tumors exhibited tumor growth inhibition following intraperitoneal BPTES administration [7], although the results were not statistically significant. Nonetheless, improving drug delivery, for example through nanoparticles, could enhance absorption and therapeutic efficacy. Pharmacokinetic and off-target limitations, as well as drug delivery methods for BPTES, should also be considered in in vivo experiments and in its potential translation to the clinic. Moreover, Metformin effects on mitochondrial activity and glucose metabolism may vary depending on dosing and patient-specific factors.

Unlike BPTES, CB-839 (telaglenastat) is a clinically optimized GLS1 inhibitor. CB-839, one of the first GLS1 inhibitors to enter advanced clinical trial [36], has entered Phase I/II clinical trials in solid tumors, including renal [63] and triple-negative breast cancers [64] in combination with CT, showing good tolerability [46]. In preclinical studies, CB-839 has also been shown to reduce growth of lymphomas, breast, renal, and pancreatic cancers.

In contrast, as a monotherapy in Phase II trials, CB-839 showed minimal antitumor activity in lung cancer [65] and resistance in pancreatic cancer [66], likely due to compensatory metabolic mechanisms in tumors. Although CB-839 is a selective GLS1 inhibitor, its activity may still be dose-dependent, and potential off-target effects cannot be fully excluded. Further investigations are needed to define its optimal therapeutic window, specificity, and safety profile.

A recent study evaluated the synergistic effect of CB-839 combined with IR on HNSCC cells [46]. CAL-27 and HN5 cell lines, when treated with standard IR, showed significantly increased GLS1 activity. However, the combination of IR with CB-839 strongly decreased cell survival, spheroid size, and tumor volume in both CAL-27 and HN5 xenograft models in athymic nude mice compared to monotherapy [46]. Furthermore, depletion of GLS1 activity in HN6 and HN31 cell lines, either through CB-839 or shRNA, led to decreased levels of glutamate, lactate, glucose, and Gln, along with reduced cell growth and increased apoptosis [38] (Figure 3). Current evidence supports that CB-839 efficacy as a single agent is limited by tumor metabolic plasticity, with its greatest therapeutic potential achieved in combination strategies. Anyway, further investigations are needed to explore and translate its potential use in HNSCC. At the clinical level, CB-839 remains the only Gln-directed agent with early-phase data available in solid tumors, a point that underscores both its translational relevance and the need for further clinical investigation in HNSCC.

### 6.2. GLS1 Enzyme Inhibition and Cancer Stemness in HNSCC

Interestingly, glutaminolysis appears to stimulate HNSCC stemness. HNSCC cancer stem cells (CSCs) exhibit high CD44 expression, an established surface marker of stem cells, and of aldehyde dehydrogenase (ALDH) [9]. ALDH belongs to the NAD(P)+-dependent enzyme family and plays a key role in oxidizing aldehydes into carboxylic acids while contributing to cancer metabolism, chemoresistance, oxidative stress resistance, and CSC survival. HNSCC CSCs show high GLS and glutamate expression, supporting their ability to form tumorspheres. Glutamate is required to switch on the active site of the ALDH. Thus, when glutamate is overexpressed due to high GLS activity, it may, in turn, regulate ALDH expression and enzymatic activity.

Inhibition of GLS1 by shRNA also suppressed CD44 expression and reduced stemness in vitro, as well as tumorigenesis in vivo. Furthermore, in a floor-of-mouth oral cancer xenograft model developed in mice (NCr-nu/nu strain) using UM-SCC-14A cells, tumors derived from tumorsphere cultures showed larger mean volumes and higher glutamate levels than those derived from adherent control cells, resulting in higher tumor incidence. Moreover, exogenous Gln supplementation promoted tumorsphere formation and increased GLS expression, further stimulating stemness via GLS [9].

### 6.3. GLS1 Inhibition and Its Impact on Mitochondrial Energetics in HNSCC

Gln is a major fuel for mitochondrial metabolism in HNSCC, replenishing the TCA cycle, supporting OXPHOS, and maintaining redox homeostasis via NADPH production. Interfering with Gln metabolism through GLS1 inhibition leads to metabolic stress that perturbs cellular redox balance and impairs mitochondrial function in HNSCC cells. As a consequence of the Warburg effect, cancer cells secrete a large amount of lactic acid into the TME. Levels can reach concentrations of up to 40 mM, compared to physiological levels of 1.5–2.5 mM in the bloodstream [67]. These elevated levels are often even higher in metastatic tumors and correlate with poor survival in HNC [42]. Lactic acid accumulation leads to an extracellular acid pH (pH 6.0–6.5), which inhibits glycolytic enzyme activity, disrupts normal biological processes, and compromises the survival of healthy cells [67]. Generally, TME acidification is primarily due to glucose-lactate conversion and protons (H^+^) release, but a significant portion of lactate is also glutaminolysis-derived. A lower pH in the TME favors cancer progression by impairing immune cell function, thus impacting responses to immunotherapy, CT and RT [42]. Nevertheless, cancer cells can quickly adapt their biochemical and energetic mechanisms to bypass these hostile conditions. Seahorse Metabolic Stress assays revealed that DON treatment in CAL33 HNSCC cells decreases basal oxygen consumption rate (OCR) and extracellular acidification rate (ECAR) reflecting impaired aerobic respiration and reduced lactate production from glutaminolysis.

Similar metabolic changes were observed in HN5 and CAL-27 cells treated with CB-839, where the OCR/ECAR ratio was significantly decreased, indicating impaired aerobic respiration [46]. In CAL-27, the dual action of CB-839 and IR increased DNA damage and oxidative stress, amplifying ROS production, thereby sensitizing cells to IR [46] (Figure 3). It is important to note that these observations are correlative, and further mechanistic studies are necessary to determine a causal relationship.

Mitochondrial energy metabolism in cancer is also affected by the dysregulation of lipoate-sensitive processes that can be targeted by CPI-613, an alpha-lipoic acid analog, that inhibits pyruvate dehydrogenase (PDH) and alpha-ketoglutarate dehydrogenase (α-KGDH). Thus, CPI-613 prevents the entry of both glucose and Gln into the TCA cycle, affecting OXPHOS, leading to decreased ATP generation, increased ROS levels, disruption of mitochondrial membrane potential, and activation of cell death pathways.

However, in HN6 and HN31 cell lines, CPI-613 triggers glutaminolysis through GLS1 upregulation, promoting tumor progression in these models. This response demonstrates the ability of tumors to reprogram their metabolism. Importantly, co-treatment with CB-839 attenuates these effects, resulting in decreased glucose consumption and lactate production otherwise induced by CPI-613. Overall, the synergistic treatment reduces ROS, suppressed cell viability, and induces apoptosis, achieving a stronger anticancer effect [38] (Figure 3).

The functional link between glutaminolysis and energy metabolism was further demonstrated by increased PDH activity following GLS1 knockdown by shRNA in HN6 and HN31 cells. This effect was abolished upon CPI-613 treatment.

In an orthotopic tongue tumor model, generated with GLS1-knockdown HN6 cells injected into immunodeficient NSG mice, tumor volume was significantly reduced, as indicated by a decrease in cells positive for the proliferation marker Ki67, without any change in the apoptotic rate. Similarly, CB-839-treated xenografts showed significantly smaller tumor volumes and lower Ki67 expression. This effect was further amplified when combined with CPI-613 due to the attenuation of compensatory glycolysis and enhancement of apoptosis along with reduced Ki67-positive cells [38].

These results reinforce a key concept: the metabolic plasticity of HNSCC may limit the effectiveness of single-agent GLS1 or mitochondrial inhibitors. Hence, a dual-target therapy may produce stronger cytotoxic effects against HNSCC cells and represents a more potent therapeutic strategy.

### 6.4. ASCT2 Transporter Inhibition in HNSCC

An alternative experimental strategy is to interfere with the Gln transporter ASCT2 in HNSCC cells to block the upstream activation of Gln metabolism by hindering its entry into the cell. Depriving cancer cells of Gln, an essential metabolic substrate, may prevent glutamate synthesis and the activation of downstream metabolic signaling, thereby impairing their ability to proliferate, evade oxidative damage, resist therapy and ultimately suppress HNSCC tumor growth.

Clinically, significantly higher serum Gln levels have been measured in HNSCC patients undergoing RT compared to those who had not received RT and non-HNSCC patients [50]. In vitro experiments conducted on murine squamous carcinoma cell lines, SCC7 and 4MOSC1 (Table 2), displayed an increase in intracellular Gln levels and a dose-dependent inhibition of cell proliferation after exposure to irradiation, along with reduced glucose uptake and glycolysis compared to untreated cells [49]. Furthermore, the TCGA dataset indicates that human HNSCC tissues are characterized by high ASCT2 mRNA levels [49,50] corroborated by protein overexpression detected through IHC, which is even higher in HPV (+) patients, and inversely associated with OS [49]. This likely explains the high serum Gln concentrations observed in patients.

High ASCT2 expression has also been linked with clinicopathological features such as poor differentiation, lymphovascular invasion, sex, tumor site, T stage, and lymph node metastasis in tongue cancer [49]. Therefore, these findings support its potential use as a marker of tumor aggressiveness in HNSCC.

Stable ASCT2 knockdown by shRNAs in human SCC15 and FaDu reduces Gln uptake and cell proliferation, with effects enhanced by the addition of V-9302 (Figure 3), an ASCT2 antagonist developed by H. Charles Manning’s team [49]. ASCT2 silencing increases ROS while simultaneously decreasing intracellular GSH, thereby making cells susceptible to apoptosis, even at low H_2_O_2_ concentrations (Figure 3). In vivo, female nude mice injected subcutaneously with ASCT2-silenced cells and treated with V-9302 developed smaller tumors confirming the in vitro results. Upregulation of ASCT2 has also been observed in post-RT HNSCC patient tissues, associated with worse survival outcomes according to IHC analysis, as well as in RT-treated SCC7 and 4MOSC1 murine cells [50], likely due to oxidative stress.

From the TCGA curated HNSCC expression profiles, it emerges that OS is prolonged in patients with low expression of Gln metabolic genes and high expression of cytotoxic T lymphocyte-related genes. Combination of RT and Gln pathway blockade with V-9302 potentiated the inhibition of cell activity in murine HNSCC cell lines by decreasing intracellular Gln levels and promoting a ferroptosis mechanism [50]. ASCT2 knockdown also improved the response of HNSCC to Cetuximab, sensitizing HNSCC to apoptosis both in vitro and in vivo [49]. Clinically, ASCT2 protein expression, assessed by IHC in HNSCC tissues, was significantly higher in Cetuximab non-responders. It is plausible to speculate that ASCT2 may play a functional role in pharmacoresistance. However, the mechanistic interplay among Gln uptake, RT or CT response, and ferroptotic cell death remains to be fully elucidated.

Nevertheless, it is still unclear whether ASCT2 overexpression actively drives malignancy in HNSCC or merely reflects downstream oncogenic signaling. Moreover, because Gln can also be transported by SNAT1 and SNAT2, ASCT2 inhibition may trigger compensatory upregulation of these alternative transporters.

A further critical concern is that normal cells rely on Gln to sustain their metabolic homeostasis, and systemic ASCT2 blockade may therefore produce toxic effects.

Hence, in light of these implications, the use of more selective and comprehensive strategies to reduce Gln entry specifically in tumor cells will be necessary prior to clinical translation and validation.

It should also be considered that combining Gln-targeting agents with RT or immunotherapy may amplify metabolic or immune-related toxicities, an aspect that preclinical models have only partially addressed so far.

### 6.5. ASCT2 Transporter Inhibition and Immune Modulation in HNSCC

In light of the evidence that ASCT2 inhibition disrupts Gln-dependent metabolic adaptation in HNSCC, combining this approach with complementary therapeutic strategies, such as immune checkpoint blockade, may further enhance anti-tumor efficacy.

Indeed, a lower pH in the TME, due to high lactate levels, impairs immune cell function, thus impacting responses to immunotherapy, CT and RT and contributing to tumor progression.

Therefore, interfering with immune response could be a promising strategy.

Song et al. demonstrated that a dual blockade of Gln uptake, by selectively targeting its transporter ASCT2, and antiphagocytic activity of CD47, through its knockdown in murine cell lines or mouse model, promotes tumor remission and enhances RT-induced ferroptosis in HNSCC [50]. As a result, CD47 has been identified as a potential immune checkpoint target therapy in human HNSCC. The CD47 protein, known as the “don’t eat me” signal, is frequently highly expressed on cancer cells and binds to signal regulatory protein alpha (SIRPα), displayed on macrophages and dendritic cells. This interaction enables immune evasion surveillance by inhibiting phagocytosis. CD47 also binds to thrombospondin-1 (TSP1), a ligand present on T cells surface, thereby inhibiting T-cell activation in the immune system [68].

The Oncomine, Cromer, and TCGA online cancer microarray databases reveal significantly increased CD47 mRNA expression or DNA copy number in most of the HNSCC datasets examined.

Moreover, IHC showed CD47 overexpression in human HNSCC tissue samples compared to normal oral mucosa, correlating with clinicopathological parameters and outcome, and conferring poor prognosis in HNSCC patients [67]. An in vivo study showed that inhibition of CD47 with monoclonal antibody in an immunocompetent transgenic HNSCC mouse model (Tgfbr1/Pten 2cKO mice) reduces tumor growth and improves TME by stimulating effector T cells and decreasing the population of immunosuppressive cells [68].

These findings were supported in C3H/He mice subcutaneously injected with SCC7 cells, where treatment with V-9302 in combination with RT significantly increased intratumoral CD8+ T cell infiltration as well as M2 macrophages. Simultaneously, CD47 expression in tumor tissues and cell lines was upregulated with a consequent reduction in macrophage phagocytosis. The mice received RT combined with either Gln blockade or anti-CD47. Anti-CD47 administration successfully reduced M2 macrophage infiltration induced by RT, improving the immunosuppressive environment and mitigating the side effects of V-9302 [49]. Similarly, in the Tgfbr1/Pten 2cKO HNSCC mice model, triple therapy achieved the greatest tumor reduction and TME improvement [50].

Therefore, triple therapy combining Gln blockade, anti-CD47, and RT improves and prolongs survival in SCC7 tumor-bearing mice, while increasing central memory T cells in the spleen [61]. The results of this multimodal approach demonstrate an enhanced anti-tumor immune response upon blocking Gln entry, supporting the hypothesis of a functional interplay between Gln metabolism and immune evasion. Therefore, the ability of ASCT2 inhibition to sensitize HNSCC to RT, CT, or immunotherapies may represent a promising therapeutic strategy to overcome immune evasion mechanisms and improve tumor control. However, the long-term effectiveness of such combination regimens will depend on the ability to prevent the emergence of new immune resistance mechanisms as tumors adapt to changes within the immune microenvironment.

Taken together, these findings indicate that GLS1 currently represents the most advanced pharmacological target, whereas ASCT2 inhibition offers a complementary strategy acting upstream of Gln catabolism. By contrast, MYC acts as an upstream regulator of both pathways. Although biologically relevant, it remains a more challenging target from a translational point of view.

## 7. Myc-Driven Regulation of Gln Metabolism in HNSCC

Metabolic rewiring is a hallmark of malignancy driven by several oncogenic signaling pathways. In many cancers, including HNSCC, genes encoding enzymes responsible for Gln metabolism are regulated by proto-oncogene c-Myc. c-Myc is a nuclear transcription factor that directly binds to the promoter region of GLS1, upregulating its transcription. c-Myc also indirectly increases GLS1 expression by repressing the microRNAs (miR-23a and miR-23b), which normally bind to the 3′-untranslated region (UTR) of GLS1 mRNA, thereby inhibiting its translation. Additionally, c-Myc promotes Gln uptake by binding to the promoter regions of the Gln transporter SLC1A5 [51,69]. Although these mechanisms establish c-Myc as a key regulator of Gln dependency in HNSCC, its subtype-specific functions and context-dependent regulatory patterns require further clarification.

IHC analysis of human HNSCC samples revealed c-Myc protein overexpression compared to normal tissues, positively correlated with tumor size, lymph node metastasis, clinical stage and associated with unfavorable clinical outcomes [47]. As indicated by Kaplan–Meier curves, c-Myc protein overexpression corresponds to poorer disease-free survival [47]. Moreover, significantly higher c-Myc mRNA levels have been observed in HPV (−) samples and patients with low c-Myc mRNA expression exhibited better OS [70]. While these correlations imply that c-Myc contributes to driving aggressive tumor phenotypes, its overexpression could also arise from upstream genomic instability or oncogenic pathway networks that are not yet fully understood. Furthermore, c-Myc contributes to an immunosuppressive TME by modulating immune infiltration. It was negatively correlated with the infiltration of B cells, CD8^+^ T cells, CD4^+^ T cells, γδ T cells, and NK cells in HNSCC, while positively correlated with neutrophils and Th17 cells. This immune profile was also associated with OS in HNSC patients [70]. Functional targeting of c-Myc using siRNA or the pharmacologic inhibitor MYCi975 (Myc inhibitor 975) in human HN6 and CAL27 cell lines impaired cell proliferation, invasion, and migration in vitro.

Furthermore, in vivo experiments using BALB/c-nude mice injected with HN6 cells and treated with MYCi975 developed smaller tumors compared with vehicle-treated controls. Mechanistically, c-Myc inhibition has been reported to induce cell apoptosis and to activate a tumor cell-intrinsic immune response via the cGAS-STING signaling pathway, promoting CD8^+^ T cell infiltration in preclinical murine models. Anyway, it is important to consider the context specificity of c-Myc–STING interactions and whether immune activation results directly from c-Myc blockade or as a consequence of other cellular stress responses. Similar results were obtained in a syngeneic murine model established by subcutaneous injection of MSCC1 cell line into immunocompetent C57BL/6 J mice. Then treatment with MYCi975 led to a reduction in c-Myc protein expression while increasing CD8^+^ T cell infiltration through the chemokine receptor CXCR3, usually highly expressed by effector T cells [47]. Accordingly, Kaplan–Meier curves showed that patients with elevated MYC expression and low CD8^+^ T cell infiltration correlated with poor OS [47].

The prevention of CD8^+^ T cell infiltration may explain the resistance observed in recurrent or metastatic HNSCC treated with PD-1–targeting immune checkpoint inhibitors combined with CT, approved as a first-line therapy. c-Myc modulated PD-L1 in esophageal squamous carcinoma as reported from [71]. In line with this, in a preclinical study, MYCi975 inhibitor, suppressed in vivo tumor growth in mice, increased tumor immune cell infiltration, upregulated PD-L1, and finally sensitized tumors to anti-PD1 immunotherapy [59].

Bioinformatic analysis of the TCGA HNSCC cohort revealed a strong correlation between c-Myc and GLS1 gene expression, suggesting the potential role of c-Myc in promoting Gln metabolism to sustain malignant progression [48]. A functional interplay between GLS1 and c-Myc is increasingly evident in HNSCC. GLS1 plays a critical role in preventing c-Myc ubiquitination through USP1 and stimulates invasion and metastasis by its interaction with the c-Myc-Slug signaling axis [48].

Treatment of human HNSCC cell lines, HN6 and HN12, with MYCi975 or c-Myc knockdown reduced cellular Gln consumption and decreased GLS1 levels. Conversely, c-Myc overexpression increased GLS1 levels in both cell lines, confirming its regulatory role in GLS1 gene expression [48]. A dual-target approach, simultaneously inhibiting GLS1 (with CB-839) and c-Myc (with MYCi975), achieved stronger suppression of the GLS1-c-Myc axis, than either agent alone in an orthotopic mouse model of HNSCC. This combined approach not only enhances tumor suppression but also inhibits metastatic progression. Consistent with genomic data, pharmacologic or genetic inhibition of c-Myc reduces Gln consumption and GLS1 expression, in contrast to the effects of its overexpression. Altogether, these results suggest that targeting oncogenic c-Myc may represent a promising therapeutic strategy in HNSCC.

## 8. TP53-Mediated Regulation of Gln Metabolism in HNSCC

The tumor suppressor gene TP53, which encodes the p53 tumor suppressor protein, is frequently mutated in HNSCC, with most mutations being of the missense type [72]. Patients with TP53 mutations were shown to have reduced OS [73,74].

In HNSCC, p53 function is often inactivated, either due to TP53 gene mutations in HPV (−) cases or via degradation mediated by the viral E6 protein in HPV (+) cases. Although HPV (+) HNSCC patients may carry either mutant or wild-type (WT) TP53 alleles, the E6 protein limits the activity of WT p53, thereby reducing its tumor suppressive functions.

In summary, HPV (+) TP53-WT HNSCC patients are characterized by significantly better survival outcomes compared to HPV (+) TP53-mutant and HPV (−) TP53-WT patients [75,76].

Mutant or dysfunctional p53 often not only loses its tumor-suppressive function but also gains oncogenic functions, triggering EMT and promoting chronic inflammation [72,77]. Additionally, these mutants can interfere with miRNAs, which are crucial in regulating genes involved in cell division, differentiation, metabolism, and malignancy. This dual loss of function/gain of function effect contributes to resistance to both RT and CT, as observed in cells with high levels of apoptosis inhibitors like Bcl2 and Bcl-XL, and low levels of the pro-apoptotic factor Bax [77].

To overcome this resistance and improve treatment efficacy, several therapeutic strategies have been developed and applied, including targeting WT p53 degradation or inhibition, reactivating its transcriptional activity by binding mutant p53, and restoring WT p53 function [77].

However, these approaches fail to address metabolic rewiring and TME alterations induced by p53 mutations in cancer. In fact, loss of p53 function has been associated with increased glycolysis, altered mitochondrial activity, and metabolic reprogramming, including enhanced dependence on exogenous nutrients such as Gln. TP53 plays a critical role in Gln metabolism by regulating the expression of glucose transporters GLUT1 and GLUT4 [41]. Based on this, it can be speculated that p53 dysfunction contributes to these metabolic alterations, while p53 induction can trigger senescence, autophagy, and apoptosis through the PI3K/AKT/mTOR and ROS signaling pathways, potentially restoring metabolic control. Some studies have reported altered GLS1 expression in association with TP53 alterations [7], suggesting that TP53-mutant tumors may be particularly dependent on Gln metabolism and potentially more sensitive to GLS1 inhibition.

p53 also upregulates the GLS2 enzyme, leading to increased GSH levels and reduced ROS levels, thereby protecting cells from DNA damage. In response to oxidative stress or DNA damage, p53 promotes GLS2 synthesis by directly binding its promoter. p53 also regulates the GLS2 and cytochrome c genes, both involved in OXPHOS [78].

Nevertheless, despite these insights, the relationship between TP53 and Gln metabolism in HNSCC remains poorly understood. Further research is needed to clarify how metabolic modulation could effectively enhance the efficacy of p53-targeted therapies.

## 9. Conclusions

HNSCC remains a challenging malignancy. Growing evidence highlights the pivotal role of Gln metabolism and the high reliance of HNSCC cancer cells on this pathway compared with normal cells. Indeed, targeting Gln metabolism represents a promising therapeutic strategy for treating aggressive forms of HNSCC.

Encouraging preclinical results in HNSCC cell lines and in vivo mouse models, which show that inhibiting Gln metabolism at key checkpoints leads to significant antitumor effects, raise concerns about their clinical translatability. Even GLS1, the most clinically advanced target, still requires further investigation and validation before being integrated into therapeutic regimens for HNSCC.

These studies present several biological and methodological limitations. HNSCCs are genetically and metabolically heterogeneous, depending on HPV status and anatomical origin. Cell line experiments fail to simultaneously capture these characteristics, and the culture conditions do not accurately mimic the physiological nutrient levels, with unstable Gln availability. More in vitro co-culture models with stromal cells, fibroblasts, and immune cells should be employed to clarify how their crosstalk influences the response to Gln-metabolism inhibitors. Similarly, in vivo xenograft models rely on immunodeficient hosts, which lack the heterogeneity, complexity, and hypoxic conditions of the human TME, limiting their predictive value. Moreover, systemic drug administration of Gln inhibitors may also be poorly tolerated, causing toxicity in the intestine, immune system, or normal tissues. Tumors may utilize host-derived Gln, thereby compromising the therapeutic efficacy of Gln-targeted strategies.

A further peculiarity of cancer cells is their remarkable ability to quickly rewire the metabolism in response to stress or the inhibition of key pathways. Upon inhibition of Gln metabolism, cancer cells may compensate by switching to alternative nutrient sources, like glucose, fatty acids or other amino acids. This adaptive plasticity enables them to survive under therapeutic pressure, contributing to treatment resistance. Furthermore, inhibiting Gln metabolism can trigger downstream stress responses, making it difficult to distinguish Gln-specific effects. Thus, there is a need to develop agents that can selectively target tumor-specific Gln dependence without affecting normal tissues and to block compensatory mechanisms. Overall, considering these challenges, more physiologically relevant models—such as patient-derived organoids and co-culture systems—are needed to better reflect the physiological conditions and improve the translation of Gln-targeted therapies [79]. Subsequently, clinical trials will be necessary to evaluate the safety and efficacy of Gln-targeting drugs in patients. Although much of the mechanistic evidence summarized in the central sections of this review is derived from cell lines and other preclinical models, their translational potential remains limited; therefore, progress in this field will depend on validating these findings in well-designed clinical studies.

Another crucial issue is the lack of predictive biomarkers to identify Gln-dependent HNSCC tumors, as Gln reliance can vary among patients. A wide biomarker panel, integrating metabolic, genetic, and TME features, may help stratify patients and optimize the selection of those who may benefit most from Gln-targeted therapeutic approaches. For example, assessing the expression of GLS1, ASCT2, or Gln could be validated as useful criteria to determine specific patient subsets. Importantly, increased GLS1 expression has been documented in multiple malignancies, such as colorectal, breast, liver, lung, prostate and intrahepatic cholangiocarcinoma, where GLS1 upregulation correlates with enhanced Gln uptake, tumor progression and poor prognosis [80].

This review examines recent studies conducted using HNSCC cell lines and patient-derived samples, highlighting the complex interplay between Gln metabolism and processes such as ferroptosis, immune evasion, mitochondrial energy production, and oncogenic transcriptional regulation. Consequently, targeting a single pathway may provide minimal benefits and limited effectiveness. Instead, a multi-targeted approach acting on different Gln pathway checkpoints, combined with pharmacologic or biological therapies that modulate other dysregulated cellular mechanisms in HNSCC, and used alongside standard RT or CT regimens, may improve therapeutic outcomes.

Furthermore, identifying specific genetic alterations and dysregulated pathways that drive tumor growth and metastasis in individual HNSCC patients, along with early diagnosis, will support the selection and implementation of appropriate combined treatments, contributing to the advancement of precision medicine in oncology.

A clinically relevant observation is that patients with HNC often exhibit Gln deficiency, and Gln supplementation has been shown to significantly reduce the incidence of chemoradiation-induced OM side effects.

From a translational perspective, the development of a multi-agent therapeutic approach could result in superior anti-HNSCC effects, fewer adverse events such as OM, and a more favorable prognosis in patients. However, the mechanism linking OM to Gln metabolism should be clarified. Identifying patients with high Gln metabolism, through potential GLS1 biomarkers or serum Gln measurement, could help to target oral Gln supplementation more effectively and improve clinical benefit. However, the full molecular mechanisms underlying HNSCC pathogenesis remain to be elucidated, and further clinical trials are essential to test and evaluate the efficacy of potential combination therapies, including those targeting Gln metabolism.

## Figures and Tables

**Figure 1 cells-14-01962-f001:**
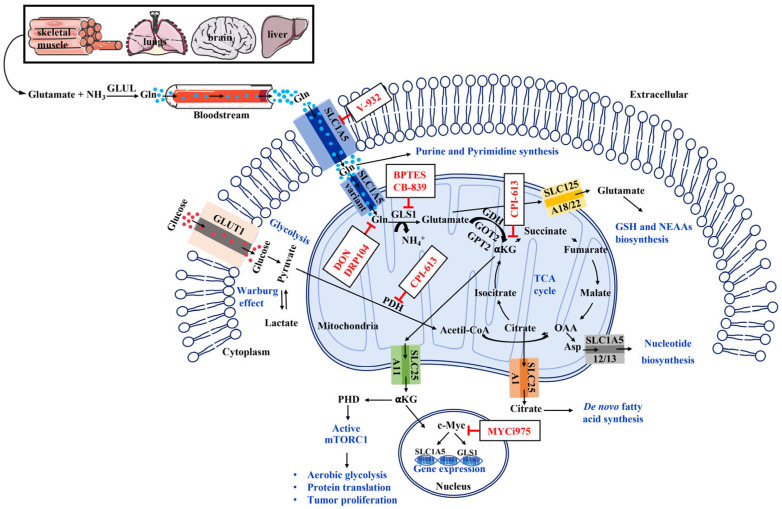
Integrated overview of Glutamine metabolism and its pharmacological inhibition in HNSCC cells. Glutamine (Gln) synthesized by GLUL in skeletal muscle, lungs, brain, and liver, is released into the bloodstream and enters cells through the SLC1A5 transporter. Cytosolic Gln is transported into mitochondria through a SLC1A5 variant and hydrolyzed to glutamate by GLS1. Glutamate is further converted to α-KG, via GDH or GOT2 and GPT2 fueling the TCA cycle and supporting energy production and biosynthetic pathways. A portion of mitochondrial glutamate is translocated to the cytosol through the SLC25A18/22 transporters, where it is recruited for GSH and NEAA biosynthesis. Cytosolic Gln also serves as a nitrogen donor in purine and pyrimidine synthesis. Citrate and α-KG can also be exported to the cytosol via SLC25A1 and SLC25A11, contributing to de novo fatty acid synthesis and mTORC1 activation, respectively. The nuclear transcription factor c-Myc enhances Gln metabolism by promoting GLS1 and SLC1A5 gene expression. Multiple pharmacological agents can interfere with this metabolic network. V-9302 blocks Gln uptake by inhibiting SLC1A5. DON and its prodrug DRP104 act as antimetabolites competing with Gln at the GLS1 active site, whereas BPTES and CB-839 selectively inhibit GLS1, blocking the conversion of Gln to glutamate. The alpha-lipoic acid analog CPI-613 targets: pyruvate dehydrogenase and alpha-ketoglutarate dehydrogenase within the TCA cycle suppressing mitochondrial metabolism. MYCi975 inhibits the oncogenic transcription factor c-Myc. Adapted from [36] and created using SMART Servier Medical Art https://smart.servier.com/SMART (accessed on 1 June 2025).

**Figure 2 cells-14-01962-f002:**
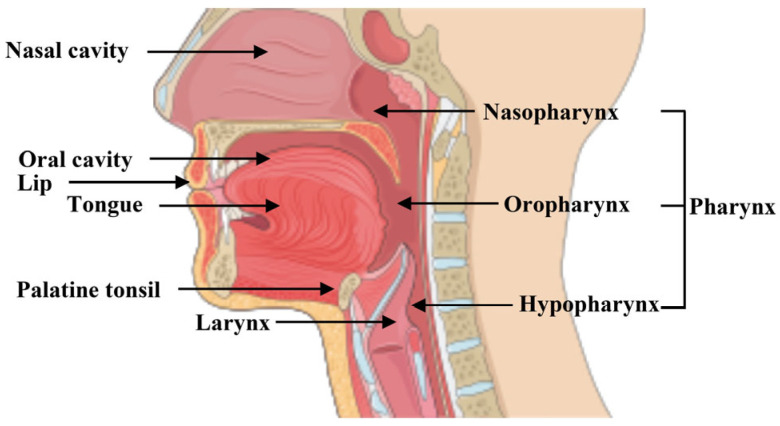
Upper aerodigestive tract. Representative sites of the upper aerodigestive tract involved in HNSCC. The most common components include: nasal cavity, oral cavity (tongue, lips, palatine tonsil), pharynx (nasopharynx, oropharynx, hypopharynx), and larynx. Figure created with SMART—Servier Medical Art https://smart.servier.com/SMART (accessed on 1 June 2025).

**Figure 3 cells-14-01962-f003:**
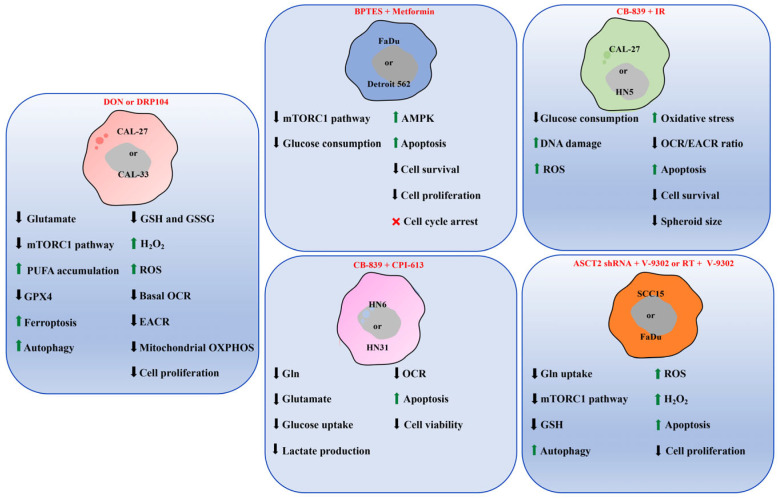
Multiple stress responses in HNSCC cell lines targeting Gln metabolism in vitro. Inhibiting glutaminolysis in HNSCC cell lines—through pharmacological or genetic inhibition of Gln uptake, GLS1 activity, or mitochondrial TCA enzymes—induces metabolic stress, including reduced mitochondrial activity (↓OCR, ↓OXPHOS), GSH depletion, impaired mTORC1 signaling, and increased ROS/H_2_O_2_ levels. These alterations promote apoptosis, autophagy, and, in some specific models, ferroptosis linked to lipid peroxidation and decreased GPX4 activity, as reported in the cited studies. Combined treatments (e.g., BPTES + metformin, CB-839 + IR, CB-839 + CPI-613, ASCT2 silencing + V-9302) have been shown to enhance these effects in vitro, reducing cell survival, glucose uptake, lactate production, and spheroid growth, thereby supporting stronger anti-tumor effects in cellular models. 
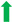
 Increase; 
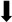
 Decrease; 
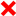
 Arrest. Figure created using SMART—Servier Medical Art https://smart.servier.com/SMART (accessed on 1 June 2025).

**Table 1 cells-14-01962-t001:** Characteristics of human and murine HNSCC cell lines.

Cell Line	Primary Site of Origin	Gender	HPV Status	GLS1 Expression	Reference
CAL-27	Tongue	Male	Negative	High	[33,47]
CAL-33	Tongue	Male		Not shown	[33]
Detroit 562	Pharynx	Female		High	[7]
FaDu	Hypopharynx	Male		High	[7,46]
HN5	Tongue	Male		High	[46]
HN6	Tongue	Male		High	[38,48]
HN12	Lymph node metastasis of the tongue	Male		High	[38,48]
HN31	Lymph node metastasis of the tongue	Male		High	[38,48]
HSC-3	Lymph node metastasis of the tongue	Male		High	[9]
OSCC-3	Tongue	Unknown		High	[9]
SCC15	Tongue	Male		Not shown	[49]
UM-SCC-14A	Floor of mouth	Female		High	[9]
UM-SCC-17B	Metastatic laryngeal cancer	Female		High	[9]
UDSCC2	Hypopharynx	Male	Positive (HPV-16)	High	[33]
UM-SCC47	Lateral tongue	Male		High	[33]
SCC7	HNSCC model established in C3H/HeJ mice		Negative	Not shown	[50]
4MOSC1/MSCC1	HNSCC model induced in C57BL/6 mice, by carcinogen 4-nitroquinoline (4NQO)				[47,50]

**Table 2 cells-14-01962-t002:** Gene and protein expression of Gln metabolism-related markers/metabolites in HNSCC patient tissue samples.

Marker	Gene Expression (TCGA Datasets, RT-qPCR)	Protein Expression (IHC)
GLS1	Upregulated in HNSCC primary and metastatic tumor tissues [7,8,49].	Overexpressed in tumor tissues compared with adjacent normal mucosa. Significantly and inversely associated with OS [9,10,49].
GLS2	Downregulated in HNSCC tissues [10,49].	Overexpressed and correlated negatively with tumor grade [10]. Its expression in tumor tissues showed a trend toward better OS [49].
SLC1A5 (ASCT2)	Upregulated and significantly associated with sex and HNSCC subtype, depending on anatomical origin [49].	Overexpressed and higher in HNSCC post-RT; higher in Cetuximab non-responder patients. Elevated expression in HPV (+) cases, correlated with poor OS. Significantly associated with advanced T stage, differentiation grade, sex, and lymph node metastasis [49,50,52].
**Metabolite**	**Metabolomic Analysis on tissues, saliva, plasma, and serum**	
Glutamate	Significantly elevated metabolite levels in primary and metastatic HNSCC samples, correlated with advanced clinicopathological features [8,9].	
Glutamine	Lower levels of Gln in metastatic HNSCC tissues compared to primary tumors, and associated with advanced stages of HNSCC. Reduced Gln concentrations also in saliva, plasma, and serum of HNSCC patients [8,9,50].	

**Table 3 cells-14-01962-t003:** Agents targeting Gln metabolism checkpoints, mitochondrial metabolism and oncogenic control in HNSCC cell lines.

Therapeutic Agent	Characteristics	Mechanism of Action	Reference
DON	Glutamine antagonist	Antimetabolites bind competitively to the active site of GLS1, preventing Gln from binding and thereby inhibiting GLS1 enzymatic activity.	[58]
DRP104			[58]
BPTES	GLS1 inhibitor	Selective inhibition of GLS1 enzyme blocks Gln conversion to glutamate.	[58]
CB-839			[46]
CPI-613	PDH/⍺KGDH inhibitor	Alpha-lipoic acid analog inhibiting tumor mitochondrial metabolism by targeting two TCA cycle enzymes: pyruvate dehydrogenase and alpha-ketoglutarate dehydrogenase, affecting cancer cell proliferation and survival.	[38]
V-9302	ASCT2 inhibitor	Inhibits of ASCT2 transporter, blocking the Gln uptake and its metabolism.	[49]
MYCi975	c-Myc inhibitor	Target the c-Myc oncogenic transcription factor.	[59]
Metformin	Antihyperglycemic drug	Inhibits mitochondrial Complex I of the electron transport chain, activates AMPK, and suppresses mTORC1 leading to reduced proliferation and altered tumor metabolism.	[60]

## Data Availability

No new data were created or analyzed in this study. Data sharing is not applicable to this article.

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
