# Peer review of "Interplay Between Glutamine Metabolism and Other Cellular Pathways: A Promising Hub in the Treatment of HNSCC"

_cells, 2025, doi:10.3390/cells14241962_

Round 1

Reviewer 1 Report

Comments and Suggestions for Authors

The authors have provided a detailed and excellent review of how pharmacological manipulating Gln metabolism, together with other radiotherapy and chemotherapy, would provide an integrated approach to treat HNSCC. The MS was well written and Tables and Figures are well presented to illustrate the main ideas of the MS. 

Then only revision needed is: to provide references in Table 1 and 3.

Reviewer 2 Report

Comments and Suggestions for Authors

This review focuses on the GLN/GLU pathway in HNSCC as important in progress of the cancer and as a potential element in treatment.  Generally, the review was informative and interesting.  However, one point remains unclear to me.  OM is mentioned in small segments throughout the paper as a consequence of radiochemotherapy and a limiting factor in treatment, with GLN supplementation aiding.  However in and around lines 115-120,  use of such  supplementation did not impact OS.  Please, explain more.  Is OM limited to a small subset of patients or, in these trials (ref. 11, 28, 29) were there alternative treatments included that did not yield OM so this was not a big enough factor to impact the effectiveness of over all treatment.  It is also likely that other cancers have similar GLS profiles, and some reference is made to this in the text.  Maybe, add a line in the discussion to elaborate on this.  While much of earlier parts of the review are based on patient trials, in the middle and end, the work shifts heavily to cell lines.  In the Conclusion (discussion), mention this and suggest how much progress we can expect from this work.

Reviewer 3 Report

Comments and Suggestions for Authors

1. A significant conceptual omission in this review is the complete absence of discussion regarding glutamate receptor (GluR) signaling, despite its well-established interaction with glutamine–glutamate metabolism and clear clinical relevance in head and neck squamous cell carcinoma (HNSCC). Glutamine catabolism yields glutamate, which not only fuels the TCA cycle but also acts as an autocrine/paracrine signaling molecule through ionotropic (NMDA, AMPA, kainate) and metabotropic (mGluR1–8) receptors. Several studies have demonstrated that mGluR4, mGluR5, and GRM3 are upregulated in oral squamous cell carcinoma and oropharyngeal cancers, correlating with tumor aggressiveness, perineural invasion, and poorer overall survival. Furthermore, NMDA receptor subunits (e.g., GRIN2A, GRIN2B) have been implicated in modulating Ca²⁺ influx, epithelial–mesenchymal transition (EMT), and cisplatin resistance in HNSCC models. By omitting this pathway, the review fails to address a key metabolic–neurotransmitter interface that directly links glutamine metabolism to tumor signaling, plasticity, and therapeutic response.

2. The review summarizes known studies but rarely engages in critical analysis or synthesis. For instance, it restates findings (e.g., GLS1 upregulation, ASCT2 expression, c-Myc correlation) without evaluating methodological quality or conflicting evidence. There is no discussion of limitations in the cited studies (e.g., in-vitro dependence on specific HNSCC lines or small-sample clinical cohorts).

3. Sections 4.1–4.5 are lengthy and highly repetitive, re-describing oxidative stress, ferroptosis, and GSH balance multiple times. The manuscript reads more like an annotated compilation of results than a logically progressive argument.

4. The title promises an integrative review (“interplay between glutamine metabolism and other pathways”), but most sections are monolithic summaries of glutamine inhibition studies—there is little true discussion of inter-pathway crosstalk (e.g., metabolic–immune, metabolic–transcriptional coupling).

5. Tables 1–4 present transcriptional and protein data from TCGA and individual studies but fail to address dataset heterogeneity, sample bias, or statistical validity. Claims such as “high GLS1 expression inversely correlates with OS” lack reference to effect size or hazard ratios.

6. The review repeatedly infers causality (“DON induces ferroptosis,” “CB-839 enhances DNA damage response”) without distinguishing between correlative and causal evidence. No mention of dose dependency, off-target effects, or pharmacokinetic limitations of inhibitors like DON, BPTES, or CB-839.

7. Although the abstract emphasizes “precision oncology,” there is no meaningful evaluation of ongoing or completed clinical trials for GLS inhibitors or Gln metabolism modulators in HNSCC. The clinical feasibility and toxicity (e.g., DON-related mucositis, metabolic syndrome) are underdeveloped.

8.  Figures 1–4 reuse Servier Medical Art schematics with minor adaptation, providing little original insight. They mainly re-illustrate basic pathways without highlighting novel interconnections specific to HNSCC. Figure 4, in particular, is overinterpreted, suggesting mechanistic certainty not supported by cited experiments.

9. Tables 1–4 are informative but verbose, containing overlapping information about cell lines, GLS1 expression, and HPV status. They could be condensed into one integrated dataset.

10. The “Conclusion” section is generic and reiterates content from the abstract without offering new insights. It fails to delineate specific research gaps, such as metabolic heterogeneity among HPV(+) vs. HPV(–) HNSCC or tumor–stroma metabolic coupling.

Comments on the Quality of English Language

Grammatically acceptable but verbose and repetitive; many sentences exceed 40 words.

Reviewer 4 Report

Comments and Suggestions for Authors

This review comprehensively examines the role of glutamine (Gln) metabolism in head and neck squamous cell carcinoma (HNSCC), focusing on its dysregulation and potential as a therapeutic target. It integrates recent evidence on the interaction between Gln metabolism and oncogenic, metabolic, and immune pathways, including MYC and TP53 regulation. The authors propose targeting glutamine metabolism checkpoints as a promising strategy for multimodal precision oncology in HNSCC.

The introduction effectively contextualises HNSCC but is overly descriptive and lengthy. It should be condensed by removing epidemiological details that are not directly linked to glutamine metabolism and instead provide a clearer rationale for why targeting Gln metabolism represents an unmet need in HNSCC treatment.

Glutamine Signalling Pathway and Its Roles in Cellular Functions: While mechanistic explanations are strong, this section lacks discussion of how normal physiological glutamine signalling differs from the tumoural state. The authors should expand on how glutamine’s metabolic flexibility contributes to cancer cell survival under stress, linking basic biochemistry to oncogenic adaptation.

Dysregulated Glutamine Metabolism in HNSCC: This section presents valuable data but lacks critical synthesis. The authors should include a more analytical discussion of causation versus correlation—whether enhanced Gln metabolism drives oncogenesis or merely reflects tumour metabolic demand—and briefly compare findings to other solid tumours.

Glutamine Metabolism Checkpoints as a Pharmacological Target (Mechanistic Focus): Descriptions of inhibitors and checkpoints are comprehensive, but the narrative reads as a catalogue. The section should integrate a critical evaluation of preclinical study limitations (e.g., small sample sizes, lack of in vivo validation) and specify which targets show realistic translational promise.

Glutamine Metabolism Checkpoints as a Pharmacological Target (Therapeutic Implications): The therapeutic implications would benefit from a clearer hierarchy of targets—prioritising GLS1, ASCT2, and MYC—and discussion of potential combinatorial toxicities when used with radiotherapy or immunotherapy. A concise summary of ongoing clinical trials would strengthen translational relevance.

MYC-Driven Regulation of Glutamine Metabolism in HNSCC: The MYC section convincingly explains regulatory mechanisms but should highlight how c-Myc modulation interacts with immune signalling beyond metabolic reprogramming. Discussion of resistance mechanisms or compensatory pathways following MYC inhibition would enhance completeness.

TP53-Mediated Regulation of Glutamine Metabolism in HNSCC: This section is underdeveloped. It needs elaboration on how TP53 status stratifies therapeutic response to GLS1 inhibitors, including the interplay between mutant TP53 gain-of-function and altered redox balance. The authors could cite emerging studies exploring p53–glutamine–ROS signalling crosstalk.

The conclusion reiterates prior points but lacks specificity. The authors should synthesise a clearer translational roadmap, outlining key unanswered questions and next steps for preclinical validation and clinical translation, rather than general statements about “multi-targeted approaches.”

Round 2

Reviewer 3 Report

Comments and Suggestions for Authors

All major concerns have been successfully addressed, with only minor opportunities for further strengthening (e.g., even more quantification in tables, deeper critique of heterogeneity).
The manuscript is now much more coherent, analytically robust, and better structured.

Reviewer 4 Report

Comments and Suggestions for Authors

Thank you for your revision